# SLIT2/ROBO1 signaling suppresses mTORC1 for organelle control and bacterial killing

Vikrant K Bhosle[1,]*, Joel MJ Tan[1,]*, Taoyingnan Li[1,2,]*, Rong Hua[1], Hyunwoo Kwon[1,3], Zhubing Li[1], Sajedabanu Patel[1], Marc Tessier-Lavigne[4,5], Lisa A Robinson[1,6,7,8], Peter K Kim[1,9], John H Brumell[1,2,6,10]

**SLIT/ROBO signaling impacts many aspects of tissue development and homeostasis, in part, through the regulation of cell growth and proliferation. Recent studies have also linked SLIT/ROBO signaling to the regulation of diverse phagocyte functions. However, the mechanisms by which SLIT/ROBO signaling acts at the nexus of cellular growth control and innate immunity remain enigmatic. Here, we show that SLIT2-mediated activation of ROBO1 leads to inhibition of mTORC1 kinase activity in macrophages, leading to dephosphorylation of its downstream targets, including transcription factor EB and ULK1. Consequently, SLIT2 augments lysosome biogenesis, potently induces autophagy, and robustly promotes the killing of bacteria within phagosomes. Concordant with these results, we demonstrate decreased lysosomal content and accumulated peroxisomes in the spinal cords of embryos from *Robo1*$^{-/-}$, *Robo2*$^{-/-}$ double knockout mice. We also show that impediment of auto/paracrine SLIT-ROBO signaling axis in cancer cells leads to hyperactivation of mTORC1 and inhibition of autophagy. Together, these findings elucidate a central role of chemorepellent SLIT2 in the regulation of mTORC1 activity with important implications for innate immunity and cancer cell survival.**

## Introduction

Macrophages are sentinel innate immune cells which constantly survey all tissues to detect the presence of microbial pathogens and their products (pathogen-associated molecular patterns; PAMPs). During immune surveillance, macrophages sample their extracellular milieu using macropinocytosis, a specialized form of endocytosis (Freeman et al, 2020). Macropinocytosis, which can be augmented by growth factor signaling, is necessary for amino acid-induced activation of mechanistic target of rapamycin complex 1 (mTORC1) in macrophages (Yoshida et al, 2015; Swanson & Yoshida, 2019). The mTORC1 activation, in turn, stimulates protein translation and inhibits catabolic processes, such as autophagy (Swanson & Yoshida, 2019). Some types of cancer cells also use macropinocytosis as a means of nutrient uptake (Commisso et al, 2013; Lee et al, 2019). Macropinocytosis is critical to promote proliferation in these cell types (Palm, 2019).

Our group recently demonstrated that Slit guidance ligand 2 (SLIT2) is an endogenous inhibitor of macropinocytosis in macrophages and cancer cells (Bhosle et al, 2020). SLIT2 is an evolutionarily conserved secreted glycoprotein which regulates directed cell migration (chemotaxis) of multiple cell types, from leukocytes to cancer cells (Wu et al, 2001; Tole et al, 2009; Tavora et al, 2020). In addition, SLIT2 controls growth and proliferation of mammalian neurons (Borrell et al, 2012; Yeh et al, 2014). SLIT2 is cleaved into bioactive N- and C-terminal fragments in vivo; with the former (N-SLIT2) binding to cell surface Roundabout receptors 1 and 2 (ROBO1-2) to induce signaling (Chedotal, 2007). Vertebrate Roundabout 3 receptor, which is expressed in macrophages (Kim et al, 2018), does not bind to SLIT proteins (Zelina et al, 2014; Jaworski et al, 2015) and Roundabout 4 receptor is exclusively expressed in endothelial cells (Huminiecki et al, 2002; Tanaka et al, 2018). We reported that SLIT2/ROBO1 signaling controls cell size and mitigates the intake of soluble bacterial PAMPs in macrophages to attenuate inflammation in mice (Bhosle et al, 2020). Our results are congruent with a growing number of studies which show that SLIT2–ROBO1 signaling has significant effects on cellular processes, other than chemotaxis (Geraldo et al, 2021; Poch et al, 2022). SLIT2 is also a known tumor suppressor and negatively regulates primary tumor growth in many types of cancers (Zhou et al, 2013; Tavora et al, 2020; Zhou et al, 2022a). In spite of the recent advances, the mechanisms responsible for SLIT2's diverse actions on innate immunity and cancer remain poorly understood so far.

[1]Cell Biology Program, The Hospital for Sick Children, Toronto, Canada   [2]Department of Molecular Genetics, University of Toronto, Toronto, Canada   [3]Department of Internal Medicine, The Ohio State University, Columbus, OH, USA   [4]Laboratory of Brain Development and Repair, Rockefeller University, New York, NY, USA   [5]Department of Biology, Stanford University, Stanford, CA, USA   [6]Institute of Medical Science, University of Toronto, Toronto, Canada   [7]Division of Nephrology, The Hospital for Sick Children, Toronto, Canada   [8]Department of Paediatrics, Faculty of Medicine, University of Toronto, Toronto, Canada   [9]Department of Biochemistry, University of Toronto, Toronto, Canada   [10]SickKids IBD Centre, Hospital for Sick Children, Toronto, Canada

Correspondence: john.brumell@sickkids.ca; pkim@sickkids.ca; lisa.robinson@sickkids.ca
*Vikrant K Bhosle, Joel MJ Tan, and Taoyingnan Li contributed equally to this work

We show here that bioactive N-SLIT2, via ROBO1-mediated activation of SRGAP2, potently inhibits mTORC1 signaling in macrophages. This results in robust changes in cell volume (size), dephosphorylation and nuclear translocation of TFEB, and subsequent transcription of TFEB-regulated genes to ameliorate lysosome biogenesis. At the same time, N-SLIT2 enhances autophagy and reduces intracellular survival of bacteria in primary murine macrophages. In line with these findings, the spinal cords of embryos from $Robo1^{-/-}$, $Robo2^{-/-}$ double KO mice have reduced lysosome content but increased the accumulation of peroxisomes. We further show that attenuation of auto/para-crine SLIT2/ROBO1 signaling in cancer cells results in hyperactivation of mTORC1 and diminished autophagic flux. Together, these findings suggest that SLIT2, a canonical neurorepellent, is a key regulator of autophagy and organelle homeostasis in innate immune and cancer cells.

## Results

### N-SLIT2-induced activation of ROBO1 inhibits mTORC1 activity in macrophages

Resting (unstimulated) macrophages are estimated to intake extracellular fluid equal to their cell volume every 4 h via macropinocytosis (Steinman et al, 1976). Activation of Roundabout 1 (ROBO1), a canonical cell surface receptor for SLIT family of chemorepellent proteins (SLIT1-3), was recently shown to suppress macropinocytosis in macrophages, thereby impairing their ability to sense specific PAMPs, and in consequence, attenuate inflammasome activation in vitro and in vivo (Bhosle et al, 2020). Because macropinocytosis is known to promote mTORC1 activation in a variety of immune cells, including macrophages, via uptake of extracellular proteins (Yoshida et al, 2015; Yoshida et al, 2018; Charpentier et al, 2020), we hypothesized that SLIT-ROBO signaling may regulate the cellular activity of this kinase complex. To test this hypothesis, we treated BMDM with a bioactive soluble fragment of the ROBO1 agonist SLIT2 (N-SLIT2) (Wang et al, 1999) and examined phosphorylation of p70S6K, a known substrate of mTORC1 (Brown et al, 1995).

After treatment with N-SLIT2, we observed a significant (Fig 1A and B) decrease in phosphorylation of p70S6K. Strikingly, p70S6K phosphorylation was unaffected by a mutant of N-SLIT2 only lacking its ROBO1-binding leucine-rich repeat D2 domain (N-SLIT2ΔD2) (Patel et al, 2012) (Fig 1A and B). The minimum dose of N-SLIT2 that showed significant mTORC1 inhibition was 30 nM (Fig 1C and D). Inhibition of mTORC1 was also observed in RAW264.7 cells (murine macrophage cell-line) but not in the mutants lacking expression of ROBO1 (Figs 1E and F and S1A). In agreement with our findings using ROBO1 KO macrophages, N-SLIT2 had no effect on phosphorylation of p70S6K in human DLD-1 adenocarcinoma cells which completely lack the N-SLIT2-binding Roundabout receptors (Zhou et al, 2011) (Fig S1A–C). Together, these findings demonstrate that N-SLIT2 treatment inhibits mTORC1 activity in macrophages through activation of ROBO1.

### N-SLIT2-induced mTORC1 inhibition is mediated by SRGAP2

After binding to N-SLIT2, ROBO1 is known to interact with members of the SLIT-ROBO family of GTPase-activating protein (GAP)

domain-containing proteins (SRGAP), which suppress the activity of P family of small GTPases (Wong et al, 2001). In particular, SRGAP2, a RAC1-specific GAP (Charrier et al, 2012), is highly expressed in macrophages and was previously shown to be necessary for SLIT2-induced cytoskeletal rearrangements in these cells (Bhosle et al, 2020). RAC1 activation is thought to play a role in actin polymerization, which in turn initiates the formation of a macropinocytic cup (Swanson, 2008; Koivusalo et al, 2010). Therefore, we asked whether SRGAP2 contributes to mTORC1 regulation. To this end, RAW264.7 cells were treated with a specific shRNA to deplete the expression of SRGAP2 (Fig S2A and B). In response to N-SLIT2, we observed that SRGAP2 knockdown (KD) prevented the loss of mTORC1 activity compared with control shRNA-treated cells (Fig 2A and B). It is noteworthy that total and phospho-p70S6K levels were increased, but did not reach statistical significance, in SRGAP2 KD macrophages (Fig 2A and B: $P$ = 0.1827 negative control shRNA Vehicle vs Srgap2 shRNA Vehicle). This could be because of very low basal SLIT-ROBO signaling in RAW264.7 cells.

Because SRGAP2 is known to promote GTP hydrolysis by RAC1 (rendering it inactive) (Charrier et al, 2012; Shin et al, 2020), we explored the impact of ROBO1 activation on RAC1 activity using a biochemical assay. We observed that cellular levels of active (GTP-bound) RAC1 were significantly decreased in response to the N-SLIT2 treatment in a manner that required SRGAP2 expression (Fig 2C). In support of our model, pharmacological inhibition of RAC1 activation by treating cells with the small molecule inhibitor, NSC23766, was sufficient to phenocopy the effect of N-SLIT2 on mTORC1 activity (Fig 2D and E).

Finally, to verify SLIT-induced changes in cellular mTORC1 activity using an independent approach, we proceeded to measure cell volume, a phenotype known to be regulated mTORC1 in mammalian cells (Fingar et al, 2002). N-SLIT2-treated macrophages exhibited an average volume loss of 5–10% (Fig S2C). The mTORC inhibitor rapamycin was used as a positive control and resulted in a 10–15% volume loss. However, N-SLIT2's effect was completely abolished in SRGAP2 KD macrophages. These findings demonstrate that SRGAP2-mediated inhibition of RAC1 contributes to suppression of mTORC1 activity after ROBO1 activation.

### N-SLIT2 induces lysosomal biogenesis by dephosphorylating TFEB

Another important cellular substrate of mTORC1, TFEB (transcription factor EB), is a master transcriptional regulator of lysosome biogenesis and autophagy (Pena-Llopis et al, 2011; Settembre et al, 2012). Phosphorylation of TFEB by mTORC1 is known to restrict its entry into the nucleus, thereby limiting its transcription factor activity. In response to nutrient restriction, mTORC1 activity is decreased, leading to TFEB dephosphorylation and translocation to the nucleus (Pena-Llopis et al, 2011).

To examine TFEB phosphorylation, we first performed immunoblotting analysis. Phosphorylated TFEB is known to undergo a mobility shift within SDS–PAGE gels, running slower (i.e., with an apparently higher molecular weight) than the dephosphorylated form (Martina & Puertollano, 2018). In RAW264.7 cells, we observed multiple TFEB bands with lower electrophoretic mobility than the non-phosphorylated protein, consistent with prior findings (Fig 3A).

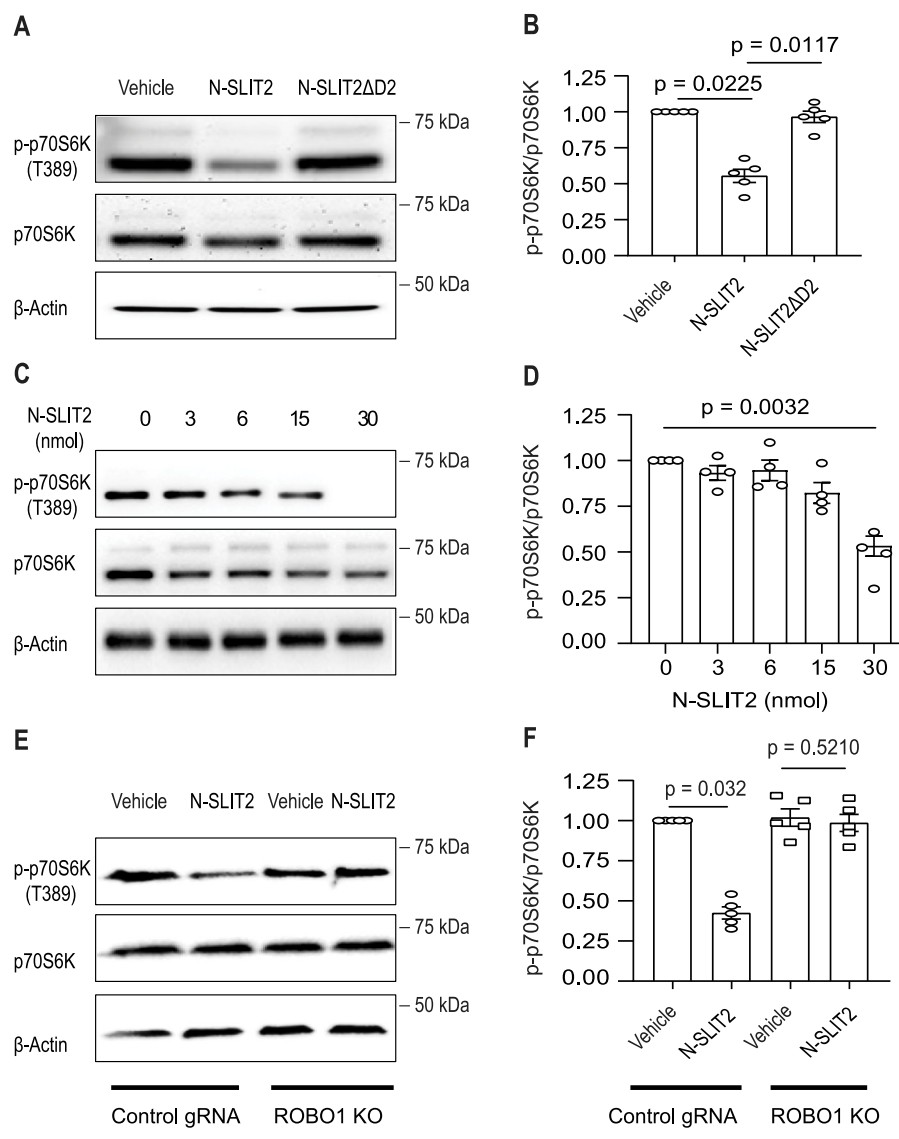

**Figure 1. N-SLIT2-induced activation of ROBO1 signaling inhibits mTORC1 activity in macrophages.**
**(A)** Murine BMDM were incubated with vehicle (DMEM), N-SLIT2 (30 nmol) or N-SLIT2ΔD2 (30 nmol) for 2 h. The p-p70s6k (phospho-T389) levels were used as a marker of cellular mTORC1 activity and β-actin was used to normalize the p-p70s6k/p70s6k ratios. **(A, B)** Quantification of p-p70s6k/p70s6k ratios normalized to vehicle control of the blots in (A). N = 5 mice. **(C, D)** Murine BMDM were treated with indicated doses of N-SLIT2 for 2 h and immunoblotted in (C) and quantified in (D). N = 4 mice. **(E, F)** RAW264.7-hCas9 cells stably transfected with non-targeting control gRNA or ROBO1 KO cells (Fig S1A) were exposed to either the vehicle or N-SLIT2 for 2 h and analyzed as in (A, B). N = 5 experimental replicates. **(B, D, F)** The graphs show mean ± SEM. Kruskal–Wallis one-way ANOVA with Dunn's Multiple Comparison Tests. Exact *P*-values are shown up to four decimal places.
Source data are available for this figure.

After SLIT2 treatment, most of the TFEB was observed in one band with higher electrophoretic mobility (Fig 3A), consistent with de-phosphorylation of the protein (Martina & Puertollano, 2018). This change of TFEB mobility was not observed in a CRISPR KO cell line lacking expression of ROBO1. Treatment of BMDM with the RAC1 inhibitor NSC23766 was sufficient to induce TFEB dephosphorylation (Fig S3), consistent with our results above. To validate our immunoblotting results, we proceeded to examine TFEB localization by immunofluorescence using antibodies directed to the endogenous protein. In control BMDM, most of the TFEB was diffusely localized to the cytosol. In response to N-SLIT2 treatment, we observed conspicuous TFEB translocation to the nucleus in most of the cells. As expected, nuclear translocation of TFEB was not observed in cells treated with bio-inactive N-SLIT2ΔD2 (Fig 3B and C). Together, these results indicate that N-SLIT2 treatment can cause TFEB dephosphorylation in macrophages.

Next, we examined the impact of N-SLIT2 on TFEB transcriptional activity and lysosome biogenesis. N-SLIT2 treatment led to elevated mRNA levels for TFEB-regulated genes, including *Flcn*, *Sqstm1*, *Gpnmb*, and *Rragd* (Fig 3D) (Carey et al, 2020). In contrast, N-SLIT2ΔD2 did not affect the transcript levels for these genes. These findings suggest that SLIT2-mediated activation of ROBO1 induced lysosomal biogenesis. Consistent with this notion, we observed elevated levels of lysosomal-associated membrane protein (LAMP1) and active cathepsin B after treatment with N-SLIT2, but not N-SLIT2ΔD2 (Fig 3E–G). Together, these findings suggest that SLIT2 activation of ROBO1 promotes lysosome biogenesis.

SLIT/ROBO signaling plays an evolutionarily conserved role to control axonal growth in the nervous system during development (Brose et al, 1999; Blockus & Chedotal, 2016). Based on our findings, we hypothesized that deficiency in SLIT/ROBO signaling would impair lysosome biogenesis in this context. To test this hypothesis, we examined spinal cords from *Robo1⁻/⁻;Robo2⁻/⁻* double KO mouse embryos (Lopez-Bendito et al, 2007). We observed decreased numbers of lysosomes (labeled with LAMP1) in these tissues compared with WT embryos (*Robo1⁺/⁺;Robo2⁺/⁺*) (Fig 3H and I). Thus, our findings

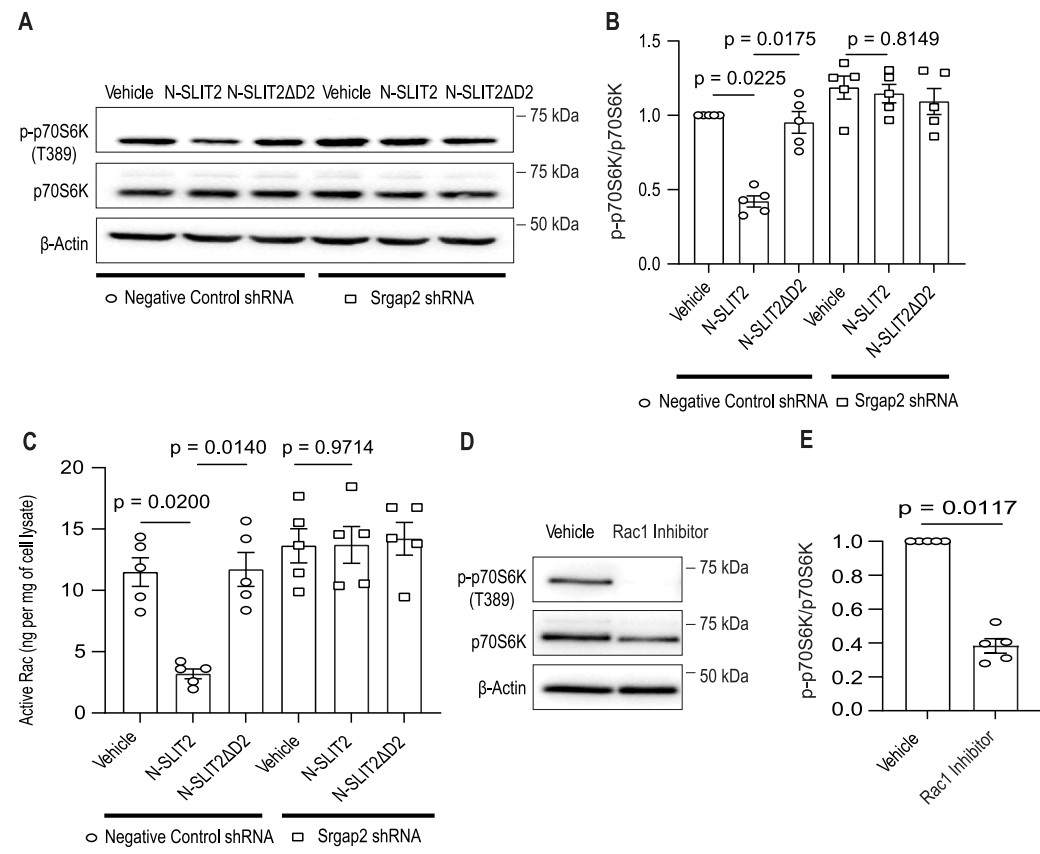

**Figure 2. N-SLIT2-induced mTORC1 inhibition is mediated by SRGAP2.**
**(A)** RAW264.7 cells stably expressing a non-targeting control shRNA or shRNA against murine *Srgap2* were treated with vehicle, N-SLIT2 or N-SLIT2ΔD2 for 2 h and lysates were analyzed by SDS–PAGE immunoblot. **(A, B)** Quantification of p-p70s6k/p70s6k ratios normalized to vehicle control of the blots in (A). N = 5 experimental replicates. **(A, C)** Cells were treated as described in (A) and active Rac levels were measured using a calorimetric G-LISA assay. N = 5 experimental replicates. **(D)** Murine BMDM were treated with either vehicle or membrane-permeable Rac1 inhibitor, NSC23766, for 2 h and experiments were performed as described in Fig 1A. **(D, E)** Quantification of p-p70S6K/p70S6K ratio normalized to vechicle in (D). N = 5 mice. **(B, C, E)** Graphs show mean ± SEM. Exact *P*-values are shown up to four decimal places. **(B, C)** Kruskal–Wallis one-way ANOVA with Dunn's Multiple Comparison Test. **(E)** Mann–Whitney *U* test.
Source data are available for this figure.

indicate that SLIT/ROBO plays a significant role in regulating lysosome biogenesis in multiple cell types.

## N-SLIT2-ROBO1 signaling promotes cellular autophagy and regulates organelle homeostasis in vivo

Inhibition of mTORC1 is known to induce autophagy through dephosphorylation of the upstream autophagy regulator, ULK1 (unc-51 like autophagy activating kinase 1), at serine 757 (Kim et al, 2011). Consistent with our findings above, we observed that phosphorylated-ULK1 levels decreased in BMDM treated with N-SLIT2 but not N-SLIT2ΔD2 (Fig 4A and B). We also observed an autophagy induction as judged by rapid loss of the autophagy adaptor, p62/SQSTM1 (Fig 4C and D) in response to N-SLIT2 but not N-SLIT2ΔD2. The loss of p62/SQSTM1 was blocked by treatment of cells with bafilomycin, an inhibitor of the vacuolar type proton ATPase (proton pump), revealing that lysosomal acidification was required. Finally, we examined autophagy by Western blotting for microtubule-associated protein 1 light chain 3 (hereafter LC3). The ratio of the lipidated (LC3-II) to the non-lipidated (LC3-I) serves as a basis to measure autophagy

induction (Klionsky et al, 2008). We observed increased ratios of LC3-II/LC3-I in response to N-SLIT2 but not N-SLIT2ΔD2, an effect that was blocked by bafilomycin treatment (Fig 4C and E). Together, these findings demonstrate that N-SLIT2 treatment induces autophagy in macrophages.

Autophagy plays a major role in organelle homeostasis (Kirkin & Rogov, 2019). Based on our findings above, we hypothesized that defects in Slit/Robo signaling would lead to autophagy deficiency during development and cause the accumulation of autophagic cargo. To test this hypothesis, we examined spinal cords from *Robo1⁻/⁻*; *Robo2⁻/⁻* (Robo1/2⁻/⁻) double KO mouse embryos (Lopez-Bendito et al, 2007). We observed increased numbers of peroxisomes (labeled with PEX14 antibodies) in these tissues compared with WT animals (Fig 4F–H). PEX14 staining was more concentrated in ventral horn regions of Robo1/2⁻/⁻ sections as compared with uniform staining in WT tissue sections (Fig 4F). Differences in LAMP1 staining were also seen in similar regions (Fig 3H). This could be because of differences in local levels of SLIT (SLIT1-3) proteins in the murine spinal cords. These findings suggest that SLIT/ROBO plays a significant role in regulating autophagy in multiple cell types.

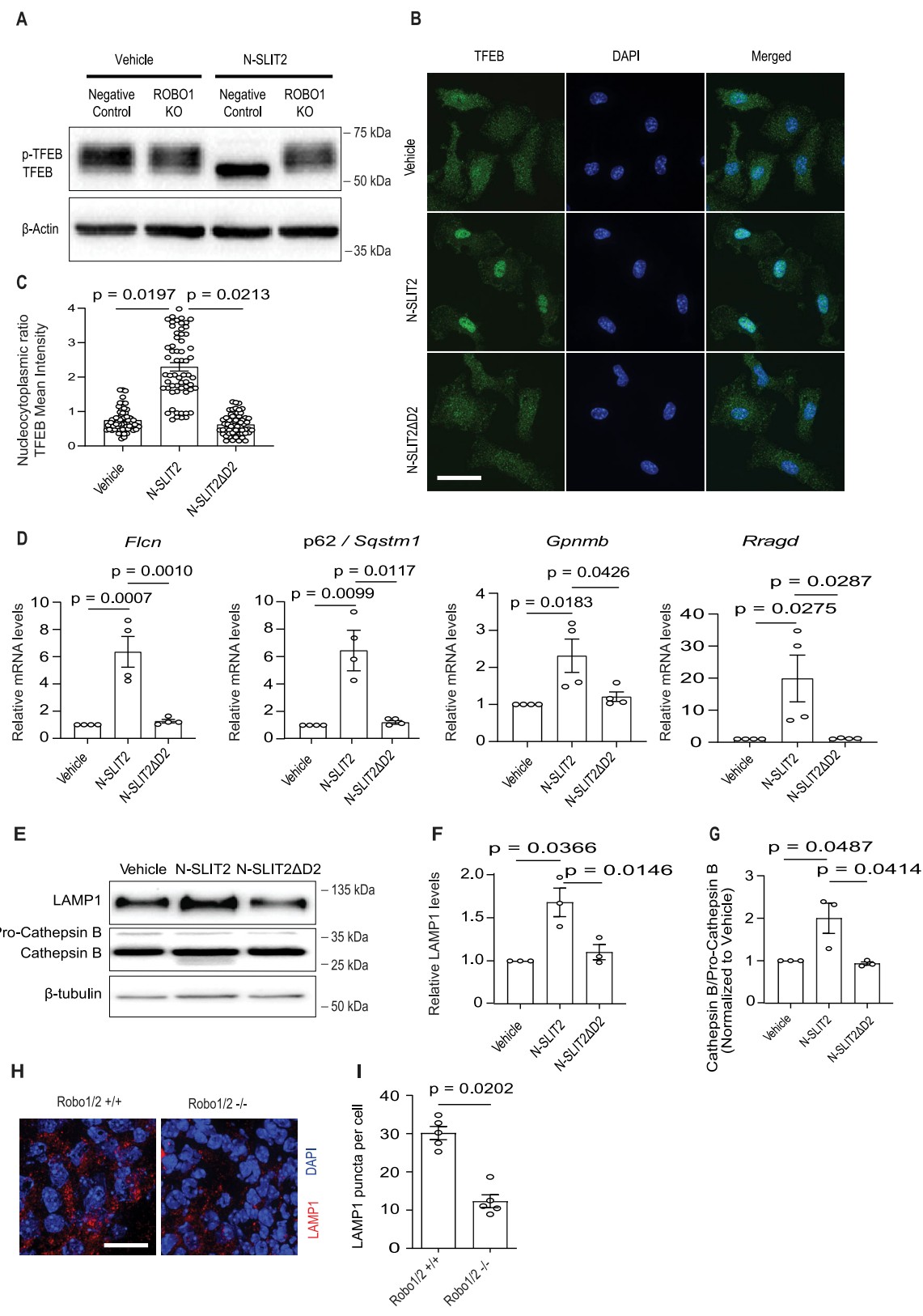

**Figure 3. N-SLIT2 induces lysosomal biogenesis by dephosphorylating TFEB.**
**(A)** RAW264.7-hCas9 cells were treated as described in Fig 1E. Cells were incubated with vehicle or N-SLIT2 for 2 h and protein lysates were immunoblotted for TFEB and β-actin (internal loading control). **(B, C)** Murine BMDM were grown on 24-well coverslips overnight and then exposed to the vehicle, N-SLIT2 or N-SLIT2ΔD2 for 2 h. The cells were fixed, permeabilized, stained for endogenous TFEB protein and nuclei (DAPI). Scale bar = 10 μm. **(B, C)** 30 cells per treatment in (B) were analyzed for TFEB localization.

### SLIT2 activation of ROBO1 promotes bacterial killing by macrophages

TFEB-mediated lysosome biogenesis (Visvikis et al, 2014) and autophagy (Kuballa et al, 2012) are known to promote killing of microbial pathogens by phagocytes. Therefore, we explored the impact of N-SLIT2 treatment on the antibacterial activity of BMDM (see experimental outline in Fig 4I). Cells were allowed to internalize two different bacteria: (i) the DH10B lab strain of *Escherichia coli*; and (ii) a triple knockout (TKO) mutant of *Listeria monocytogenes* (lacking phagosome escape factors LLO, PI-PLC, and PC-PLC) that is incapable of escaping the phagosome. After uptake, the cells were washed and the cell impermeant antibiotic gentamicin was added to the medium to kill extracellular bacteria. Cells were lysed at different times post internalization, and intracellular viable bacteria were then plated at different times post internalization. We observed faster clearance of intracellular *E. coli* (Fig 4J) and *L. monocytogenes* TKO (Fig 4K) in cells treated with N-SLIT2 but not N-SLIT2ΔD2. These results demonstrate that N-SLIT2-induced activation of ROBO1 signaling enhances bacterial killing in phagosomes by macrophages.

### High basal SLIT/ROBO signaling represses mTORC1 activity in HeLa cells

It is known that some cell lines (e.g., HeLa) have high expression of cell surface ROBO1 and secrete SLIT2 under normal growth conditions (Stella et al, 2009). Based on our findings above, we hypothesized that basal SLIT2/ROBO1 signaling would repress mTORC1 activity under normal growth conditions. To test this hypothesis, we knocked down the expression of *ROBO1* in HeLa cells using siRNA (Fig S4A). We observed that KD of *ROBO1* led to (i) increased mTORC1 activity (Fig S4B and C), (ii) increased cell size (Fig S4D and E), (iii) accumulation of peroxisomes (Fig S4F and G), (iv) a decrease in autophagic flux (Fig S4H and I), (v) a decrease in xenophagic flux (as judged by retention of LC3 colocalization with invasive *Salmonella*) (Fig S4J), and (vi) an increase in intracellular growth of these bacteria (Fig S4K). Together, these findings reveal that auto/paracrine SLIT2/ROBO1 signaling in HeLa cells impacts organelle control and innate immune defenses to bacterial infection via repression of mTORC1 under normal growth conditions.

## Discussion

Present findings provide further insight into the mechanisms by which SLIT2/ROBO1 signaling impacts innate immunity. We propose a model (Fig S5) whereby N-SLIT2-induced activation of ROBO1 inhibits RAC1 activity via the ROBO1-interacting GAP protein, SRGAP2. Inactivation of RAC1 leads to inhibition of macropinocytosis (West et al, 2000), which is required to sustain mTORC1 activity in macrophages (Yoshida et al, 2015; Yoshida et al, 2018). In addition to immune surveillance, macropinocytosis plays an essential role in nutrient uptake by immune cells (macrophages and T cells), which in turn, augments cellular mTORC1 activity (Yoshida et al, 2015; Charpentier et al, 2020). Thus, by limiting nutrient uptake via macropinocytosis, ROBO1 activation inhibits mTORC1 activity and dephosphorylation of its downstream targets p70S6K, TFEB, and ULK1. In this manner, ROBO1 activation promotes lysosome biogenesis and autophagy and engenders macrophages with an enhanced ability to kill internalized pathogens.

Several recent studies have implicated SLIT2-ROBO1 signaling in the regulation of macrophage polarization (Ahirwar et al, 2021; Geraldo et al, 2021). In addition, Wang et al used Slit2-Tg (SLIT2 transgenic mice) mice to report that the transgenic mice show increased pro-inflammatory gene expression (M1-like phenotype) in periodontitis as compared with the WT counterparts (Wang et al, 2020). Full-length SLIT2 protein is cleaved into N- and C-terminal fragments in vivo (Nguyen Ba-Charvet et al, 2001). It has been recently demonstrated that C-SLIT2 binds to its own receptor (not ROBO1/2) to induce signaling (Delloye-Bourgeois et al, 2015; Svensson et al, 2016). Detailed studies will be needed to fully elucidate the role of SLIT2 signaling in inflammation (and macrophage polarization) in future.

We show that spinal cord sections from Robo1/2$^{-/-}$ KO mouse embryos exhibit reduced number of lysosomes and impaired pexophagy (peroxisomal autophagy). Our results are in line with previous studies reporting that SLIT-ROBO signaling negatively regulates cell proliferation, a phenotype also known to be regulated by mTORC1 signaling in nonimmune macropinocytic cells, including primary neurons (Borrell et al, 2012; Yeh et al, 2014). Together, these findings elucidate an explicit role of SLIT2-ROBO1 signaling in the regulation of cellular mTORC1 activity in both immune (macrophages) and nonimmune (neurons) cells. It has been reported that Robo1/2$^{-/-}$ and Slit1/2$^{-/-}$ mouse interneurons show distinct morphological features, that is, increased size and branching, but the underlying mechanisms had remained poorly understood so far (Kimura et al, 2007). In agreement with our results, Gan et al recently reported that Srgpa2 KD activates mTORC1 signaling in neuronal cells (Gan et al, 2023). Our results provide further insight that SLIT-ROBO signaling could be involved in neurite retraction via inhibition of mTORC1, and consequent activation of autophagy.

We show that SLIT2/ROBO1 signaling can impact mTORC1 activity in two ways: (i) in a rapid, inducible manner, through addition of exogenous ligand, N-SLIT2, to macrophages which do not secrete SLIT proteins at a resting state (Dun et al, 2019; Wang et al, 2021), and (ii) in a constitutive manner, through basal auto/paracrine

N = 3 mice. **(D)** BMDM were treated with vehicle, N-SLIT2 or N-SLIT2ΔD2 for 8 h. The mRNA expression levels of 4 lysosomal–resident proteins (*Flcn*, *Sqstm1*, *Gpnmb*, and *Rragd*) were determined using qRT-PCR. N = 4 mice. **(E, F, G)** BMDM were exposed to vehicle, N-SLIT2 or N-SLIT2ΔD2 overnight (18 h) and lysates were immunoblotted for LAMP1 and cathepsin B. β-tubulin was used as internal loading control. LAMP1 (F) and cathepsin B (G) protein levels were quantified and normalized to the vehicle treatment. N = 3 mice. **(H, I)** E11.5 old Robo1/2 −/− or WT (Robo1/2 +/+) mouse embryos. The sections were stained for lysosomal marker (LAMP1) and nuclei (DAPI). **(I)** The number of LAMP1 punta per cell was calculated in (I). N = 4 mice per genotype group. Scale bar = 100 μm. All graphs show mean ± SEM. Exact *P*-values are shown up to four decimal places. **(C, D, F, G)** One-way ANOVA with Tukey's multiple comparison tests. **(I)** Mann–Whitney *U* test.
Source data are available for this figure.

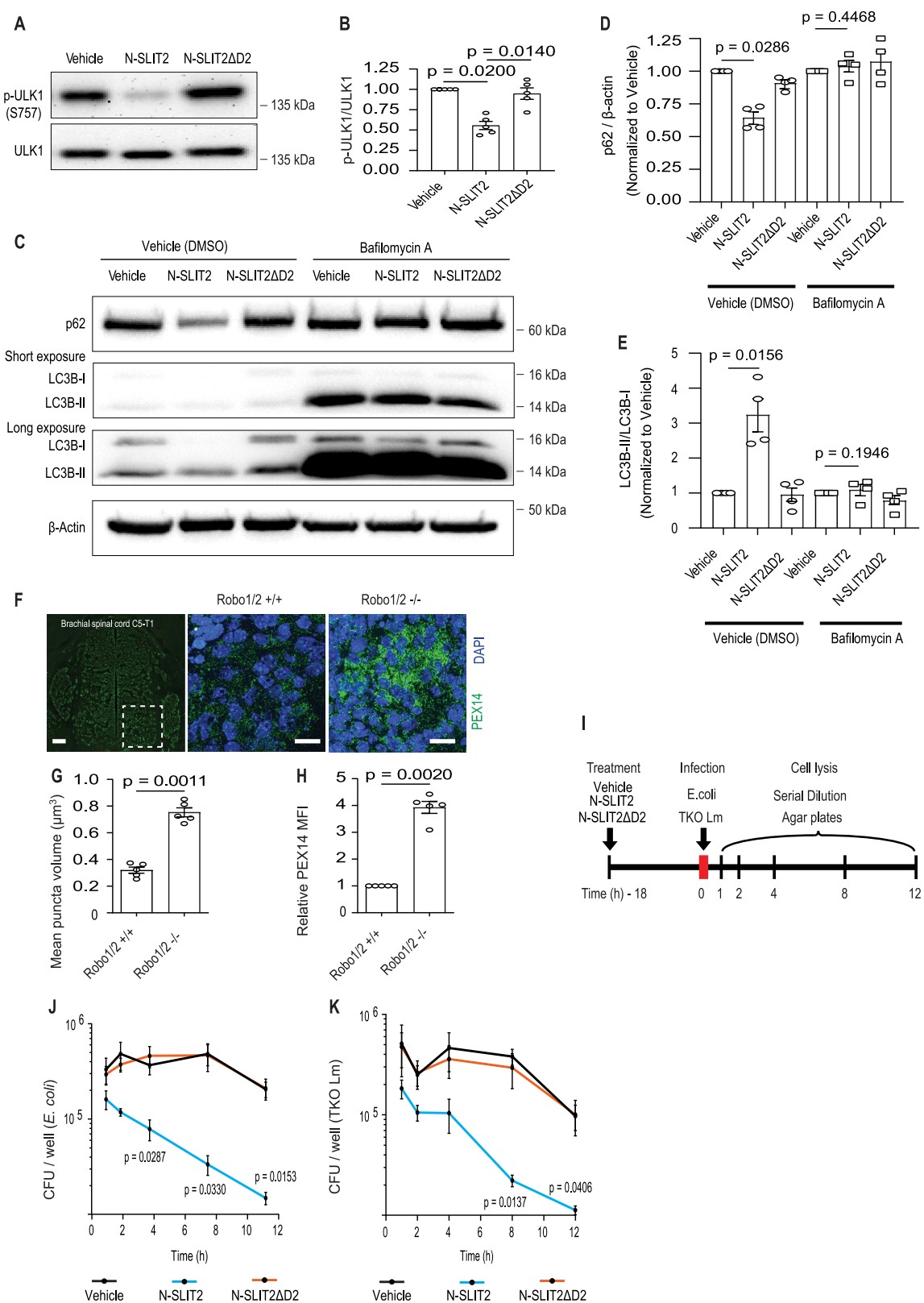

**Figure 4. N-SLIT2-ROBO1 signaling promotes cellular autophagy and regulates organelle homeostasis in vivo.**
**(A, B)** Murine BMDM were treated with vehicle, N-SLIT2or N-SLIT2ΔD2 for 2 h. **(B)** The p-ULK1 (phospho-S[757]) levels were used as a marker of cellular mTORC1 activity and quantified in (B). N = 5 mice. **(C, D, E)** Murine BMDM were incubated with vehicle (DMSO) or Bafilomycin A for 2 h and with vehicle, N-SLIT2 or N-SLIT2ΔD2 for an additional 8 h. The lysates were immunoblotted for p62 (SQSTM1) and LC3B. β-actin was used as internal loading control. The p62/β-actin (D) and LC3B-II/LC3B-I ratios (E) were

signaling in cervical cancer cells (HeLa) which express basal levels of SLIT2 and ROBO1 (Stella et al, 2009; Bianchi et al, 2021). We speculate that HeLa cells may have adapted to genomic and transcriptomic alterations (Landry et al, 2013) by fine-tuning of endogenous SLIT2/ROBO1 signaling, to modulate constitutive mTORC1 activity and maintain a delicate balance between cell proliferation, basal autophagy, and lysosome biogenesis. Recently, inactivating mutations in SLIT2 and ROBO1 were identified in a subset of relapsing cholangiocarcinoma patients (Zhou et al, 2022a). In striking concordance with our results using ROBO1 siRNA in HeLa cells, the inactivating mutations resulted in hyperactivation of mTORC1 signaling in cancer cells (Zhou et al, 2022a; Zhou et al, 2022b). Thus, mTORC1 regulation by SLIT/ROBO signaling may play an important role in cancer cell adaptation to genomic alterations. It should be noted that mTORC1 activity is not entirely inhibited by any of the SLIT2 treatments shown here. Indeed, HeLa cells can grow rapidly despite basal SLIT2/ROBO1 signaling under normal growth conditions. This suggests that SLIT2/ROBO1 signaling may serve to "titrate" the activity of mTORC1 or control its activity within specific subcellular compartments. Together, our findings reveal a central mechanism regulating both developmental pathways and innate immunity and may provide new therapeutic targets for diseases involving altered SLIT/ROBO signaling.

# Materials and Methods

### Mice

Animal studies were reviewed and approved by The Hospital for Sick Children (SickKids) (protocol #1000052111 J.H.B.) and The Centre for Phenogenomics (protocol #21-0326 L.A.R.) animal care committees in accordance with the guidelines established by Canadian Council on Animal Care. 8–12 wk old C57BL/6 mice (Charles River Laboratories) were used for isolation of BMDM as described below. $Robo1^{+/-}$; $Robo2^{+/-}$ mice (Lopez-Bendito et al, 2007) were bred in-house to produce $Robo1/2^{+/+}$ (WT) and $Robo1/2^{-/-}$ (double KO) embryos. The genotyping was performed as previously described (Grieshammer et al, 2004; Long et al, 2004).

### Cell culture and primary murine BMDM isolation

All cell lines used in the study were authenticated and tested negative for mycoplasma by SickKids Biobank and the Mycoplasma Plus PCR kit (Agilent Technologies), respectively. RAW264.7 (TIB-71) cells were purchased from American Type Culture Collection. The cells were cultured in (DMEM; Hyclone Laboratories) supplemented with 10%

heat-inactivated (FBS; Wisent) without antibiotics at 37°C and 5% $CO_2$. Murine BMDM were obtained from the dissected femurs and tibias of 8–12-wk old C57BL/6 mice. The cells were washed with growth medium and plated on 70 $cm^2$ Petri dishes (Thermo Fisher Scientific). The medium was replaced every 2–3 d, and after 7 d, the cells were used for experiments. BMDM were maintained in high-glucose RPMI-1640 medium (#350-025-CL; Wisent) containing 10% heat-inactivated FBS (Wisent), 25 ng/ml murine M-CSF (PeproTech), 1% penicillin and streptomycin (Thermo Fisher Scientific). HEK293-EBNA1-6E cells (a gift from Dr. Yves Durocher, Montreal, QC, Canada) were cultured in Freestyle F17 culture medium (Thermo Fisher Scientific) and were used for N-SLIT2ΔD2 production. The hCas9-expressing RAW264.7 (RAW264.7-hCas9) cells (Napier & Monack, 2017) were a kind gift from Dr. Denise M Monack (Stanford University, Stanford, CA, USA) and were cultured in DMEM with 10% heat-inactivated FBS, and 10 μg/ml blasticidin (Sigma-Aldrich) (blasticidin selection media). HEK293T (CRL-3216) and HeLa (CCL-2) cells were purchased from American Type Culture Collection and were cultured in DMEM with 10% heat-inactivated FBS, and 1% streptomycin–penicillin.

### Bacterial strains

*L. monocytogenes* (*Lm*) was grown overnight in brain–heart infusion broth and the DP-L2319 (Δ*hly*Δ*plcA*Δ*plcB*, TKO) strain was used (Tan et al, 2021). TKO *Lm* was sub-cultured until reaching an $OD_{600}$ of 0.3 before infection. Overnight culture of *E. coli* was grown in lysogeny broth and the DH10B strain was used. *E. coli* was cultured overnight, diluted to an $OD_{600}$ of 1.0, and opsonized with IgG (Meridian Bioscience) for 30 min at 37°C before the infection. *Salmonella enterica* serovar Typhimurium were grown in lysogeny broth and the SL1344 strain was used.

### Antibodies and reagents

All antibodies used in this study are listed in Table S1. Recombinant human N-SLIT2 was purchased from PeproTech. In N-SLIT2ΔD2, ROBO1/2-binding domain was removed from N-SLIT2 (aa 235–444), replaced with a short linker (Patel et al, 2012), and cloned into a pTT28 vector (a gift from Dr. Yves Durocher, Université de Montréal, QC, Canada) which contains a C-terminal (His)6x tag between the NheI and BamHI restriction sites (Hu et al, 2018). Large-scale production of N-SLIT2ΔD2 was performed in HEK293-EBNA1-6E cells, as described previously (Tole et al, 2009). All recombinant proteins were tested for endotoxin contamination using ToxinSensorTM Chromogenic LAL Endotoxin Assay Kit (GenScript). The endotoxin levels were less than 0.05 EU/ml in all recombinant protein preparations. Final concentrations of 30 nM N-SLIT2 or

---

calculated. N = 4 mice. **(F, G, H)** E11.5 old Robo1/2 −/− or WT (Robo1/2 +/+) mouse embryos. The box focuses on the ventral horn neurons. Peroxisomes were stained using antibody against PEX14 and peroxisomal volume and mean PEX14 intensity per cell were calculated. Scale bar = 100 μm. SLIT2 activation of ROBO1 promotes bacterial killing by macrophages. **(I)** Schematic representation of BMDM treatments. The cells were first treated with vehicle, N-SLIT2 or N-SLIT2ΔD2 for 18 h. **(J, K)** On the next day, the cells were infected with either (J) *E. coli* or (K) TKO Lm in the presence of a fresh round of vehicle, N-SLIT2, or N-SLIT2ΔD2 treatments for indicated times. At the end of each treatment, the cells were lysed and bacterial CFUs were counted by serial dilution. **(B, D, E, G, H, J, K)** Graphs show mean ± SEM. Exact *P*-values are shown up to four decimal places. **(B, D, E)** Kruskal–Wallis one-way ANOVA with Dunn's multiple comparison test. **(G, H)** Mann–Whitney *U* test. **(J, K)** One-way ANOVA with Tukey's multiple comparison tests.
Source data are available for this figure.

N-SLIT2ΔD2 were used for all experiments, unless indicated otherwise. Bafilomycin A (#11038; 100 nM) was from Cayman Chemical.

## Plasmids

The ptfLC3 (GFP-RFP-LC3) plasmid was a gift from Tamotsu Yoshimori (Kimura et al, 2007) (plasmid # 21074; Addgene; http://n2t.net/addgene:21074; RRID: Addgene_21074). The following plasmids were a gift from Didier Trono (Dull et al, 1998).

1 pMDLg/pRRE (plasmid #12251; Addgene; http://n2t.net/addgene:12251; RRID: Addgene_12251).
2 pMD2.G (plasmid # 12259; Addgene; http://n2t.net/addgene:12259; RRID: Addgene_12259).
3 pRSV-Rev (plasmid # 12253; Addgene; http://n2t.net/addgene:12253; RRID: Addgene_12253).

## Bacteria intracellular killing/growth assays

BMDMs were plated at $3 \times 10^5$ cells per well in 24-well tissue culture plates and treated with the vehicle (DMEM), N-SLIT2 (30 nM) or N-SLIT2ΔD2 (30 nM) for 18 h before infection. The cells were infected with TKO *Lm* or *E. coli* at an MOI of 1 or 1:200, respectively, in RPMI. At 60 min postinfection (p.i.), cells were washed three times with PBS+/+ (Wisent) and cultured in RPMI medium containing 10% FBS, 50 $\mu$g ml$^{-1}$ gentamicin (#400-130-IG; Wisent). At 1, 2, 4, 8, and 12 h p.i., the cells were lysed with PBS+ containing 1% Triton X-100. Serial dilutions of the lysates were plated on LB (*E. coli*) and brain-heart infusion (TKO *Lm*) agar plates and incubated at 37°C for 16 h for subsequent quantification of intracellular CFUs.

To examine intracellular growth of *Salmonella*, a previously established approach was used for infection of epithelial cells using late-log bacterial cultures as inocula (Steele-Mortimer et al, 1999). Briefly, bacteria were pelleted at 10,000 x g for 2 min and resuspended in PBS, pH 7.2. The bacteria were diluted and added to the cells at 37°C for 10 min. For invasion experiments, bacteria were diluted 1:50 in PBS.

## Transfections and RNA interference

For siRNA-mediated KD, HeLa cells were seeded in 12-well tissue culture plates at a concentration of $3 \times 10^4$ cells/well 24 h before use. The cells were then transfected with siRNA at a final concentration of 50 nM using Lipofectamine RNAiMax (Thermo Fisher Scientific) for 72 h as recommended by the manufacturer. Human ROBO1 silencer select validated siRNA (#s12092; sense 5′–3′ sequence- GAUCAUACCUUGAAAAUUAtt) was purchased from Ambion (Life Technologies). Non-targeting control siRNA (#D-001210-01; 5′–3′ sequence- UAGCGACUAAACACAUCAA) was purchased from Dharmacon (GE Healthcare). The siRNA against ATG12 (GUGGGCA-GUAGAGCGAACA) was custom synthesized from MilliporeSigma.

## Lentiviral particle production

The lentiviral protocol (#1000047152) was reviewed and approved by the SickKids Biosafety Committee. The following plasmids were purchased from Sigma-Aldrich (MISSION shRNA).

1 Non-targeting shRNA vector: SHC002 (Fritz et al, 2015).
2 Murine Srgap2 shRNA (Fritz et al, 2015): TRCN0000271904, sequence: 5′-CCGGCATGACCTGTCTGAT- ATTATTCTCGAGAATAATATCAGA CAGGTCATGTTTTTG-3′.

80% confluent HEK293T cells (in a 15-cm tissue culture dish) were incubated with 2 $\mu$g of pLKO.1-puro plasmid (Sigma-Aldrich), 7.5 $\mu$g pRSV-Rev (Addgene), 7.5 $\mu$g pMDLg/pRRE (Addgene), 5 $\mu$g pMD2.G (Addgene), and 88 $\mu$l PEI prime reagent (#919012; Sigma-Aldrich) diluted in 15 ml of Opti-MEM (Gibco™, Thermo Fisher Scientific). The medium was changed to antibiotic-free DMEM with 10% heat-inactivated FBS after every 24 h. Media containing the shRNA-packaged virus particles were collected at 48, 72, 96 h, and spun down for 5 min at 524$g$, filtered (0.45 $\mu$m filter), and supplemented with polybrene (Sigma-Aldrich) (10 $\mu$g/ml) before adding to the RAW264.7 cells. 72 h after adding lentiviral particles, cells were selected with 5 $\mu$g/ml puromycin for additional 2–3 wk before confirming the gene KD using Western blotting.

## Generation of *Robo1* KO macrophage cell-line

RAW264.7-hCas9 murine macrophage cells, stably expressing hCas9 (Napier et al, 2016), were used to generate a Robo1 KO macrophage cell line. The following plasmids were purchased from Sigma-Aldrich (MISSION CRISPR gRNA) in LV2 vector format.

1. Negative control gRNA: sequence (5′-3′): CGCGATAGCGCGAA TATATT.
2. Mouse Robo1 gRNA: MMPD0000034074: sequence (5′-3′): GCAC CGCATCGACCCCCAG.

Lentiviral particles were produced as described above, and were added to RAW264.7-hCas9 cells (Napier & Monack, 2017). Transfected cells were selected by adding 5 $\mu$g/ml of puromycin (Sigma-Aldrich) to the blasticidin selection media after 72 h. Single control and Robo1 KO colonies were selected by serial dilution and KO was confirmed using Western blotting.

## Drug treatment

For RAC1 inhibition, BMDM were serum-starved for 2 h, followed by treatment with 50 $\mu$M cell-permeable RAC1 inhibitor (#553502; Sigma-Aldrich) for an additional 2 h. The cellular Rac activity was measured using the G-LISA Rac Activation Assay kit (#BK125; Cytoskeleton, Inc.), as per the manufacturer's instructions. For mTORC1 inhibition, cells were serum-starved for 2 h and then treated with 100 nM rapamycin for the indicated times.

## Cell volume measurements

Cells were serum-starved for 2 h and then treated with vehicle, N-SLIT2 (30 nM) or rapamycin (100 nM) overnight (24 h) as indicated. Volume measurements were performed as previously described with some modifications (Freeman et al, 2020). Cells were gently lifted by scraping and mean cell diameter (>10,000 cells) was measured using the Z2 Coulter particle count and size analyzer (Beckman Coulter). Mean diameter was converted to volume by assuming that the suspended cells were spherical.

## Immunofluorescence

For all fixed microscopy-based experiments, cells were fixed with 2.5% PFA (#15710; Electron Microscopy Sciences) in PBS for 10 min at 37°C, unless indicated otherwise. Immunostaining was performed as previously described (Tan et al, 2021).

## Confocal microscopy

Unless otherwise indicated, cells were imaged using a Quorum spinning disk microscope with a 63x, 1.4 NA oil immersion objective (Leica DMIRE2 inverted fluorescence microscope equipped with either a Hamamatsu Back-Thinned EM-CCD camera or Hamamatsu CMOS FL-400 camera) and Volocity 6.3 software (PerkinElmer). Confocal $z$-stacks of 0.3 $\mu$m were acquired. Images were analyzed with the Volocity software and then imported into and assembled in Adobe Illustrator for labelling.

## Immunoblotting

For all immunoblotting experiments, cells were grown in six-well plates and used at 75% confluency for treatments. After the indicated treatments, the cells were washed with ice-cold PBS and lysed using RIPA lysis buffer, supplemented with protease–phosphatase inhibitor (MS-SAFE; Sigma-Aldrich). Protein concentration was determined using Bio-Rad DC protein assay (Bio-Rad laboratories). 30 $\mu$g cellular protein lysates were resolved by 12% SDS–PAGE, transferred to PVDF membrane (Bio-Rad), and probed with antigen-specific primary antibodies. Blocking was performed with 5% skim milk; except for the phospho antibodies, which was instead performed with 5% bovine serum albumin. For all analyses, HRP-conjugated secondary antibodies were used (peroxidase-conjugated goat anti-rabbit IgG or peroxidase-conjugated goat anti-mouse IgG (Jackson ImmunoResearch)) and detection was performed using SuperSignal West Femto Maximum Sensitivity Substrate (Thermo Fisher Scientific). The results were analyzed using Image Lab v6.1 by Bio-Rad laboratories.

## Embryo spinal cord sections and tissue staining

Mouse embryos were dissected from pregnant dams at day E13.5 post-conception. The tissue was then placed in OCT medium (#4583, Tissue-Tek; Sakura Finetek) and snap-frozen using liquid nitrogen. 5-$\mu$m sections collected in four series were obtained using a Cryostat (Leica CM1850; Leica Biosystems) and mounted on charged slides (#CA48311-703; VWR International). The tissue was fixed in ice-cold methanol for 10 min at –20°C. The slides were blocked with 5% BSA, 10% goat serum, and 0.1% Triton X-100 diluted in PBS- for 30 min at RT. Tissue were stained with antibodies to LAMP1 and PEX14 overnight at 4°C, washed with PBS and counterstained with secondary antibodies and DAPI. Tissues were imaged on a Quorum spinning disk confocal scanning microscope (Quorum Technologies Inc.) equipped with x10 and x60 objectives. LAMP1 and PEX14 puncta and fluorescence intensities were quantified using Volocity 6.3 software as previously described (Tan et al, 2018).

## Quantitative RT–PCR (qRT–PCR)

Total RNA was extracted from cells using the RNeasy Mini Kit (#74104; QIAGEN). cDNA was synthesized from 1 μg total RNA by reverse transcription-polymerase chain reaction (RT–PCR) using the iScript Reverse Transcription Supermix (#1708841; Bio-Rad laboratories). Quantitative real-time PCR (qRT-PCR) was performed using ssoFast EvaGreen Supermix (#1725201; Bio-Rad laboratories) with the following primers according to the manufacturer's instructions. The following cycling parameters were used: 95°C for 30 s, then 40 cycles of 95°C for 5 s, 60°C for 20 s, and the melt curve stage is 95°C for 15 s, 65°C for 1 min, and ramp up to 95°C with a rate of 0.15°C/s.

## Statistics

Statistical analyses were conducted using GraphPad Prism v9.0. Shapiro–Wilk normality test was used to verify Gaussian (normal) distribution. The means ± S.E.M. are shown in the figures. The $P$-values (reported up to four decimal points) were calculated using either $t$ test or one-way ANOVA (parametric) or Kruskal–Wallis one-way ANOVA (nonparametric) or Mann–Whitney $U$ test. Tukey's HSD test was used as a post-hoc test for one-way ANOVA (parametric), and Dunn's test was used as a post-hoc test for Kruskal–Wallis one-way ANOVA, as indicated in the figure legends. A $P$-value of less than 0.05 was considered statistically significant.

# Data Availability

The source data underlying main figures are provided as a source data file. All other data that support the findings are available from the corresponding authors upon reasonable request.

**Primers' table.**

| Mouse gene | Forward primer (5′-3′) | Reverse primer (5′-3′) |
|---|---|---|
| Gpnmb | TGCCAAGCGATTTCGTGATGT | GCCACGTAATTGGTTGTGCTC |
| Sqstm1 | GAGGCACCCCGAAACATGG | ACTTATAGCGAGTTCCCACCA |
| Flcn | AACGCCATAGTCGCCCTCT | GCTGCTCATCTGAATGCCAC |
| Rragd | AGGAGCGGCAAGTCGTCTAT | CCGGCAGATCCTGTTGGTG |
| Hprt | CTGGTGAAAAGGACCTCTCGAAG | CCAGTTTCACTAATGACACAAACG |

# Supplementary Information

# Acknowledgements

This work was supported by operating grants from the Canadian Institutes of Health Research to LA Robinson (PJT-169167), PK Kim (PJT-156196), and JH Brumell (FDN #154329). JH Brumell holds the Pitblado Chair in Cell Biology. Infrastructure for the Brumell Laboratory was provided by a John Evans Leadership Fund grant from the Canadian Foundation for Innovation and the Ontario Innovation Trust. LA Robinson holds a Canada Research Chair in vascular inflammation and kidney injury. We thank Denise Monack (Stanford University) for RAW cells expressing Cas9 (RAW264.7-hCas9). We thank Dr. Spencer Freeman for scientific discussions.

## Author Contributions

VK Bhosle: formal analysis, investigation, methodology, and writing—original draft, review, and editing.
JMJ Tan: formal analysis, investigation, methodology, and writing—original draft.
T Li: formal analysis, investigation, visualization, methodology, and writing—original draft, review, and editing.
R Hua: investigation and methodology.
H Kwon: conceptualization, investigation, methodology, and writing—original draft.
Z Li: resources and methodology.
S Patel: resources.
M Tessier-Lavigne: resources and writing—original draft.
LA Robinson: conceptualization, resources, supervision, funding acquisition, methodology, and writing—original draft.
PK Kim: resources, supervision, funding acquisition, and writing—original draft.
JH Brumell: conceptualization, resources, data curation, supervision, funding acquisition, project administration, writing—original draft, review, and editing.

## Conflict of Interest Statement

The authors declare that they have no conflict of interest.

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
