## [Reviewer comments · Life Science Alliance]

Life Science Alliance

SLIT2/ROBO1 signaling suppresses mTORC1 for organelle control and bacterial killing

Vikrant Bhosle, Joel Tan, Taoyingnan Li, Rong Hua, Hyunwoo Kwon, Zhubing Li, Sajadebanu Patel, Marc Tessier-Lavigne, Lisa Robinson, Peter Kim, and John Brumell

DOI: <https://doi.org/10.26508/lsa.202301964>

Corresponding author(s): John Brumell, Hospital for Sick Children

Review Timeline:

Submission Date:	2023-02-01
Editorial Decision:	2023-03-14
Revision Received:	2023-05-26
Editorial Decision:	2023-05-30
Revision Received:	2023-05-31
Accepted:	2023-05-31

Transaction Report:

March 14, 2023

Re: Life Science Alliance manuscript #LSA-2023-01964-T

Dr. John H Brumell
Hospital for Sick Children
Cell Biology Program
686 Bay Street, PGCRL
Toronto, ON M5G 1X8
Canada

Dear Dr. Brumell,

Thank you for submitting your manuscript entitled "SLIT2/ROBO1 signaling suppresses mTORC1 for organelle control and bacterial killing" to Life Science Alliance. The manuscript was assessed by expert reviewers, whose comments are appended to this letter. We invite you to submit a revised manuscript addressing the Reviewer comments.

Thank you for this interesting contribution to Life Science Alliance. We are looking forward to receiving your revised manuscript.

Sincerely,

B. MANUSCRIPT ORGANIZATION AND FORMATTING:

Reviewer #1 (Comments to the Authors (Required)):

This interesting manuscript extends prior studies of SLIT2/ROBO1 signaling beyond macropinocytosis, RhoA and M1 macrophage polarization to studies of autophagy that impact organelle abundance. In the main the design is thorough and the data are convincing and support the presented conclusions. I have two suggestions.

First, as Wang et al (2020 Front. Cell Dev Biol) show that SLIT2/ROBO1 signaling induces M1 macrophage polarization and cytokines in periodontitis, whether this is also true for BMDMs in this study should be shown or at least commented on in the Discussion. Second, the choice of Pex14 as a peroxisome marker is suboptimal in this context. Pex2 and/or Pex5 are preferred as Pex2 is both essential for pexophagy and a target of mTORC1, whereas Pex5 is a key substrate of Pex2 E3 ubiquitin ligase activity. Showing these data would enhance the manuscript and reinforce the conclusions, but are not essential.

Reviewer #2 (Comments to the Authors (Required)):

Comments on Bhosle et al.,

In this interesting manuscript, the authors document a new role for Slit-Robo signaling in regulating phagocytosis, lysosome biogenesis and autophagy in macrophages. Previous work from these groups implicated Slit-Robo signaling in the control of micropinocytosis in macrophages, and the present study builds on this work. Overall, the data presented are generally of high quality and strongly support the authors' central conclusions. Nevertheless, there are several areas that require further clarification that could improve the present manuscript.

Major comment

The data presented for the role of Robo1/2 in the control of lysosome biogenesis in the developing spinal cord is not particularly convincing, nor in the opinion of this reviewer is it a particularly important finding in the broader context of the study. For some reason the authors focus their analysis on the ventral spinal cord (maybe the motor column, though it is not clearly delineated in the text or figure. Based on my reading of the literature, Robo1 and Robo 2 protein expression are predominantly observed in the dorsally located commissural neurons, and the proteins (at least Robo1) are mainly observed in the post-crossing commissural axons. It is unclear if the cells that the authors are showing even express Robo proteins (mRNA has been observed in these regions, so maybe they express very low levels?). If the authors more carefully document Robo protein expression in the cells that they see changes in lysosomes and peroxisomes, this would be more convincing. Alternatively, (in my opinion) the authors could remove this data without altering the major conclusions.

Minor Comment

In Figure 1 the authors propose that Slit induces a concentration dependent decrease in p706K. It is not clear if the statistical comparisons presented really support this, rather than a binary of Slit decrease phosphorylation. Were the different concentrations significantly different from each other?

Responses to Reviewers: Bhosle et al. Manuscript #LSA-2023-01964-T

We sincerely thank both reviewers for taking the time to evaluate our manuscript and for their helpful suggestions to improve it. **Please note that author responses are in the red font below.**

Reviewer #1 (Comments to the Authors (Required)):

This interesting manuscript extends prior studies of SLIT2/ROBO1 signaling beyond macropinocytosis, RhoA and M1 macrophage polarization to studies of autophagy that impact organelle abundance. In the main the design is thorough and the data are convincing and support the presented conclusions.

Response: We thank the reviewer for their positive assessment of our work and for conducive suggestions below.

I have two suggestions.

First, as Wang et al (2020 Front. Cell Dev Biol) show that SLIT2/ROBO1 signaling induces M1 macrophage polarization and cytokines in periodontitis, whether this is also true for BMDMs in this study should be shown or at least commented on in the Discussion.

Response: We thank the reviewer for bringing the study to our attention. Macrophage polarization induced by SLIT2-ROBO1 signaling is an active topic of research. Geraldo *et al.* recently demonstrated that N-SLIT2 treatment up-regulates anti-inflammatory gene expression (M2-like) in BMDMs (Geraldo LH et al, 2021). On the other hand, Ganju and colleagues have shown that N-SLIT2 induces M1-like phenotype in tumor-associated macrophages (TAMs) and BMDMs (Ahirwar DK et al, 2021, Kaul K et al, 2021). In the study pointed out by the reviewer, Wang et al. used Slit2-Tg (genetic overexpression of SLIT2) mice to report that the transgenic mice show increased pro-inflammatory gene expression (M1-like phenotype) in periodontitis as compared to the wild type counterparts. Full-length SLIT2 protein is cleaved into N- and C-terminal fragments *in vivo* (Nguyen Ba-Charvet KT et al, 2001). While N-SLIT2 binds to the Roundabout receptors (ROBO1 because ROBO2 is not expressed in macrophages), C-SLIT2 binds to its own receptor (not ROBO1/2) to induce signaling (Delloye-Bourgeois C et al, 2015, Svensson KJ et al, 2016). We hypothesize that cellular signaling induced by SLIT2 in inflammation might be context-dependent and will be carefully investigated in future studies. We now elaborate on this important point in revised Discussion.

Second, the choice of Pex14 as a peroxisome marker is suboptimal in this context. Pex2 and/or Pex5 are preferred as Pex2 is both essential for pexophagy and a target of mTORC1, whereas Pex5 is a key substrate of Pex2 E3 ubiquitin ligase activity. Showing these data would enhance the manuscript and reinforce the conclusions, but are not essential.

Response: Anti-Catalase, Anti-PMP70 and Anti-PEX14 antibodies are the most commonly used antibodies for peroxisomes due to their specificity and fidelity in labeling peroxisomes for immunofluorescence analysis. However, we found that only the PEX14 antibody produced high quality

images for immunofluorescence microscopy of mouse embryo dorsal horn neurons. Currently, all commercially available antibodies for PEX2 are not sufficient for immunofluorescence microscopy of tissues. We have previously demonstrated that PEX2 is highly unstable as they are rapidly degraded to prevent pexophagy (Sargent G et al, 2016, van Zutphen T et al, 2016), and require 40 ug of total protein to detect in a western blot. The commercial antibody for PEX5 is excellent for almost all immunotechniques. However, the majority of PEX5 is cytosolic as it is the cytosolic receptor for peroxisomal matrix protein. We (Demers ND et al, 2023) and others have shown that only a small fraction of PEX5 is on peroxisomes. Therefore, it was not appropriate for the quantification of peroxisomes. For these reasons we feel strongly that PEX14 is the best marker to visualize and quantify peroxisomes in mouse embryo dorsal horn neurons.

Reviewer #2 (Comments to the Authors (Required)):

Comments on Bhosle et al.,

In this interesting manuscript, the authors document a new role for Slit-Robo signaling in regulating phagocytosis, lysosome biogenesis and autophagy in macrophages. Previous work from these groups implicated Slit-Robo signaling in the control of micropinocytosis in macrophages, and the present study builds on this work. Overall, the data presented are generally of high quality and strongly support the authors' central conclusions. Nevertheless, there are several areas that require further clarification that could improve the present manuscript.

Response: We thank the reviewer for overall assessment and constructive feedback.

Major comment

The data presented for the role of Robo1/2 in the control of lysosome biogenesis in the developing spinal cord is not particularly convincing, nor in the opinion of this reviewer is it a particularly important finding in the broader context of the study. For some reason the authors focus their analysis on the ventral spinal cord (maybe the motor column, though it is not clearly delineated in the text or figure. Based on my reading of the literature, Robo1 and Robo2 protein expression are predominantly observed in the dorsally located commissural neurons, and the proteins (at least Robo1) are mainly observed in the post-crossing commissural axons. It is unclear if the cells that the authors are showing even express Robo proteins (mRNA has been observed in these regions, so maybe they express very low levels?). If the authors more carefully document Robo protein expression in the cells that they see changes in lysosomes and peroxisomes, this would be more convincing. Alternatively, (in my opinion) the authors could remove this data without altering the major conclusions.

Response: We thank the reviewer for raising this important point. SLIT2 (and ROBO proteins) was first identified as an evolutionarily conserved guidance cue in the nervous system (Wang KH et al, 1999). We now elaborate more on the receptor expression in the spinal column in the revised manuscript. We have now clarified the region of interest (as the reviewer correctly pointed out “motor neurons”) in the figure legend (Fig. 4F). SLIT2-binding Roundabout receptors (ROBO1/2) are ubiquitously expressed throughout the spinal cord (including the ventral motor column) after embryonic day-11.5 (E11.5) in mice (Kim M et

al, 2017, Yuan W et al, 1999). The vertebrate ROBO3 doesn't bind to bioactive N-terminal fragments of SLIT proteins, including N-SLIT2 (Zelina P et al, 2014) and ROBO4 is exclusively expressed in endothelial cells. We have added this explanation in revised Introduction.

Minor Comment

In Figure 1 the authors propose that Slit induces a concentration dependent decrease in p70S6. It is not clear if the statistical comparisons presented really support this, rather than a binary of Slit decrease phosphorylation. Were the different concentrations significantly different from each other?

Response: We have now changed the wording to clarify that 30 nM of N-SLIT2 was the dose which showed the decrease in cellular mTORC1 activity in BMDMs (see page #4). In the absence of N-SLIT2, molecular interaction between ROBO1 C-terminus and SRGAP2 is partially inhibited by SH3 domain of the latter (Guez-Haddad J et al, 2015). Upon N-SLIT2 binding to ROBO1 (at its extracellular Ig1 domain), multiple RhoGAP effectors, including SRGAP2, are recruited to and cooperatively interact with ROBO1 C-terminus. The lack of response at lower doses of N-SLIT2 (less than 30 nM) could be due to weak intermolecular interaction between ROBO1 and SRGAP2.

References:

- Ahirwar DK, Charan M, Mishra S, Verma AK, Shilo K, Ramaswamy B, Ganju RK (2021) Slit2 inhibits breast cancer metastasis by activating m1-like phagocytic and antifibrotic macrophages. *Cancer Res* 81: 5255-5267. doi:10.1158/0008-5472.CAN-20-3909
- Delloye-Bourgeois C, Jacquier A, Charoy C, Reynaud F, Nawabi H, Thoinet K, Kindbeiter K, Yoshida Y, Zagar Y, Kong Y, et al (2015) Plexina1 is a new slit receptor and mediates axon guidance function of slit c-terminal fragments. *Nat Neurosci* 18: 36-45. doi:10.1038/nn.3893
- Demers ND, Riccio V, Jo DS, Bhandari S, Law KB, Liao W, Kim C, McQuibban GA, Choe SK, Cho DH, et al (2023) Pex13 prevents pexophagy by regulating ubiquitinated pex5 and peroxisomal ros. *Autophagy*: 1-22. doi:10.1080/15548627.2022.2160566
- Geraldo LH, Xu Y, Jacob L, Pibouin-Fragner L, Rao R, Maissa N, Verreault M, Lemaire N, Knosp C, Lesaffre C, et al (2021) Slit2/robo signaling in tumor-associated microglia and macrophages drives glioblastoma immunosuppression and vascular dysmorphia. *J Clin Invest* 131: doi:10.1172/JCI141083
- Guez-Haddad J, Sporny M, Sasson Y, Gevorkyan-Airapetov L, Lahav-Mankovski N, Margulies D, Radzimanowski J, Opatowsky Y (2015) The neuronal migration factor srgap2 achieves specificity in ligand binding through a two-component molecular mechanism. *Structure* 23: 1989-2000. doi:10.1016/j.str.2015.08.009
- Kaul K, Benez M, Mishra S, Ahirwar DK, Yadav M, Stanford KI, Jacob NK, Denko NC, Ganju RK (2021) Slit2-mediated metabolic reprogramming in bone marrow-derived macrophages enhances antitumor immunity. *Front Immunol* 12: 753477. doi:10.3389/fimmu.2021.753477

- Kim M, Fontelonga TM, Lee CH, Barnum SJ, Mastick GS (2017) Motor axons are guided to exit points in the spinal cord by slit and netrin signals. *Dev Biol* 432: 178-191. doi:10.1016/j.ydbio.2017.09.038
- Nguyen Ba-Charvet KT, Brose K, Ma L, Wang KH, Marillat V, Sotelo C, Tessier-Lavigne M, Chedotal A (2001) Diversity and specificity of actions of slit2 proteolytic fragments in axon guidance. *J Neurosci* 21: 4281-4289.
- Sargent G, van Zutphen T, Shatseva T, Zhang L, Di Giovanni V, Bandsma R, Kim PK (2016) Pex2 is the e3 ubiquitin ligase required for pexophagy during starvation. *J Cell Biol* 214: 677-690. doi:10.1083/jcb.201511034
- Svensson KJ, Long JZ, Jedrychowski MP, Cohen P, Lo JC, Serag S, Kir S, Shinoda K, Tartaglia JA, Rao RR, et al (2016) A secreted slit2 fragment regulates adipose tissue thermogenesis and metabolic function. *Cell Metab* 23: 454-466. doi:10.1016/j.cmet.2016.01.008
- van Zutphen T, Ciapaite J, Bloks VW, Ackereley C, Gerding A, Jurdzinski A, de Moraes RA, Zhang L, Wolters JC, Bischoff R, et al (2016) Malnutrition-associated liver steatosis and atp depletion is caused by peroxisomal and mitochondrial dysfunction. *J Hepatol* 65: 1198-1208. doi:10.1016/j.jhep.2016.05.046
- Wang KH, Brose K, Arnott D, Kidd T, Goodman CS, Henzel W, Tessier-Lavigne M (1999) Biochemical purification of a mammalian slit protein as a positive regulator of sensory axon elongation and branching. *Cell* 96: 771-784.
- Yuan W, Zhou L, Chen JH, Wu JY, Rao Y, Ornitz DM (1999) The mouse slit family: Secreted ligands for robo expressed in patterns that suggest a role in morphogenesis and axon guidance. *Dev Biol* 212: 290-306. doi:10.1006/dbio.1999.9371
- Zelina P, Blockus H, Zagar Y, Peres A, Friocourt F, Wu Z, Rama N, Fouquet C, Hohenester E, Tessier-Lavigne M, et al (2014) Signaling switch of the axon guidance receptor robo3 during vertebrate evolution. *Neuron* 84: 1258-1272. doi:10.1016/j.neuron.2014.11.004

May 30, 2023

RE: Life Science Alliance Manuscript #LSA-2023-01964-TR

Dr. John H Brumell
Hospital for Sick Children
Cell Biology Program
686 Bay Street, PGCRL
Toronto, ON M5G 1X8
Canada

Dear Dr. Brumell,

Thank you for submitting your revised manuscript entitled "SLIT2/ROBO1 signaling suppresses mTORC1 for organelle control and bacterial killing". We would be happy to publish your paper in Life Science Alliance pending final revisions necessary to meet our formatting guidelines.

- please add a Category for your manuscript in our system
- please ensure that the names of your co-authors are spelled correctly (names in the system and the manuscript text should match)
- please add an Author Contributions section to your main manuscript text
- please add a conflict of interest statement to your main manuscript text
- please add your main and supplementary figure legends to the main manuscript text after the references section
- please add a callout for Figure S4J to your main manuscript text
- Only one panel is present in Figure S3, so it is unnecessary to label it as "a" Please remove the label from the figure and its legend

Figure checks:

- the top 2 blots in Figure S1b look strange, while the source data images look fine. Maybe they were compressed too much for the paper image, but please re-do.
- you may consider uploading Supplemental Figure 5 as a Graphical Abstract instead, but this is up to you

A. FINAL FILES:

B. MANUSCRIPT ORGANIZATION AND FORMATTING:

Sincerely,

May 31, 2023

RE: Life Science Alliance Manuscript #LSA-2023-01964-TRR

Dr. John H Brumell
Hospital for Sick Children
Cell Biology Program
686 Bay Street, PGCRL
Toronto, ON M5G 1X8
Canada

Dear Dr. Brumell,

Thank you for submitting your Research Article entitled "SLIT2/ROBO1 signaling suppresses mTORC1 for organelle control and bacterial killing". It is a pleasure to let you know that your manuscript is now accepted for publication in Life Science Alliance. Congratulations on this interesting work.

DISTRIBUTION OF MATERIALS:

Again, congratulations on a very nice paper. I hope you found the review process to be constructive and are pleased with how the manuscript was handled editorially. We look forward to future exciting submissions from your lab.

Sincerely,
